# Instruction-driven history-aware policies
# for robotic manipulations

**Pierre-Louis Guhur**[1], **Shizhe Chen**[1], **Ricardo Garcia**[1],
**Makarand Tapaswi**[2], **Ivan Laptev**[1], **Cordelia Schmid**[1]
[1]Inria, École normale supérieure, CNRS, PSL Research University [2]IIIT Hyderabad
https://guhur.github.io/hiveformer/

**Abstract:** In human environments, robots are expected to accomplish a variety of manipulation tasks given simple natural language instructions. Yet, robotic manipulation is extremely challenging as it requires fine-grained motor control, long-term memory as well as generalization to previously unseen tasks and environments. To address these challenges, we propose a unified transformer-based approach that takes into account multiple inputs. In particular, our transformer architecture integrates (i) natural language instructions and (ii) multi-view scene observations while (iii) keeping track of the full history of observations and actions. Such an approach enables learning dependencies between history and instructions and improves manipulation precision using multiple views. We evaluate our method on the challenging RLBench benchmark and on a real-world robot. Notably, our approach scales to 74 diverse RLBench tasks and outperforms the state of the art. We also address instruction-conditioned tasks and demonstrate excellent generalization to previously unseen variations.

**Keywords:** Robotics Manipulation, Language Instruction, Transformer

## 1 Introduction

People can naturally follow language instructions and manipulate objects to accomplish a wide range of tasks from cooking to assembly and repair. It is also easy to generalize to new tasks by building upon skills learned from previously seen tasks. Hence, one of the long-term goals for robotics is to create generic instruction-following agents that can generalize to multiple tasks and environments.

Thanks to significant advances in learning generic representations for vision and language [2, 3, 4, 5], recent work has made great progress towards this goal [6, 7, 8, 9, 10]. For example, CLIPort [9] exploits CLIP models [5] to encode single-step visual observations and language instructions and to learn a single policy for 10 simulated tasks. BC-Z [10] uses a pre-trained sentence encoder [11] to generalize to multiple manipulation tasks. However, several challenges remain underexplored. One important challenge is that sequential tasks require to track object states that may be hidden from current observations, or to remember previously executed actions. This behaviour is hard to model with recent methods that mainly rely on current observations [9, 10].

Another challenge concerns manipulation tasks that require precise control of the robot end-effector to reach target locations. Such tasks can be difficult to solve with single-view approaches [12], especially in situations with visual occlusions and objects of different sizes, *e.g.* see *put money in safe* Figure 1 (left). While several recent approaches combine views from multiple cameras by converting multi-view images into a unified 2D/3D space [13, 14] or through a late fusion of multi-view predictions [15], learning representations for multiple camera views is an open research problem. Furthermore, cross-modal alignment between vision, action, and text is challenging, in particular when training and test tasks differ in terms of objects and the order of actions, see Figure 1 (right). Most of existing methods [9, 10, 16, 17] condense instructions into a global vector to condition policies [18] and are prone to lose fine-grained information about different objects.

To address the above challenges, we introduce *Hiveformer* - a **H**istory-aware **i**nstruction-conditioned multi-**vie**w trans**former**. It converts instructions into language tokens given a pre-trained language

6th Conference on Robot Learning (CoRL 2022), Auckland, New Zealand.

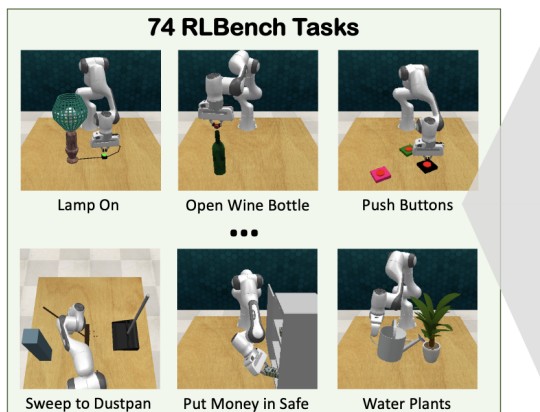
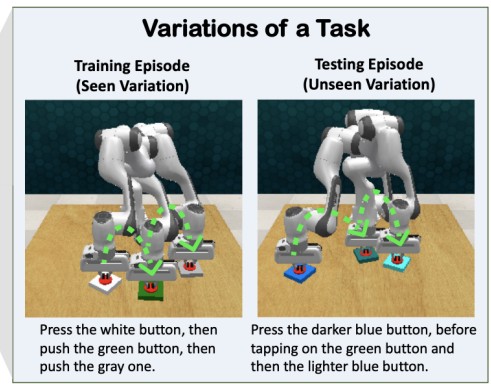

**Figure 1: Left:** Hiveformer can adapt to perform 74 tasks from RLBench [19] given language instructions. **Right:** Multiple variations of the *push buttons* task.

encoder [5], and combines visual tokens for both past and current visual observations and proprioception. These tokens are concatenated and fed into a multimodal transformer which jointly models dependencies between the current and past observations, spatial relations among views from multiple cameras, as well as fine-grained cross-modal alignment between vision and instruction. Based on the output representations from our multimodal transformer, we predict 7-DoF actions, *i.e.*, position, rotation and state of the gripper, with a UNet [20] decoder.

We carry out extensive experiments on RLBench [19] in three setups: single-task learning, multi-task learning, and multi-variation generalization[1]. Our Hiveformer significantly outperforms state-of-the-art models for all three settings, demonstrating the effectiveness of encoding instruction, history and views from multiple cameras with the proposed transformer. Moreover, we evaluate our model on 74 tasks of RLBench, which goes beyond the 10 tasks used by Liu *et al*. [15]. We manually group all the tasks into 9 categories according to their main challenges and analyze results per category for a better understanding. Hiveformer not only excels in the multiple task setting with seen instructions in training, but also enables generalization to new instructions that represent different variations of the task, even with human-written language instructions. Finally, we evaluate our model deployed on a real robot and show excellent performance. Interestingly, pretraining the model in the RLBench simulator results in significant performance gains when only a small number of real robot demonstrations are available.

To summarize, our contributions are three-fold:
- We introduce a new model, Hiveformer, to solve various challenges in robotics tasks. It jointly models an instruction, multiple views, and history via a multimodal transformer for action prediction in robotic manipulation.
- We perform extensive ablations of our model on RLBench with 74 tasks grouped into 9 distinct categories. The history improves long-term tasks and the multi-view setting is most helpful for tasks requiring high precision or in the presence of visual occlusions.
- We demonstrate that Hiveformer outperforms the state of the art in three RLBench setups, namely single-task, multi-task and multi-variation. A single Hiveformer trained with synthetic instructions is able to solve multiple tasks and task variations, can generalize to unseen human-written instructions and shows excellent performance on a real robot after finetuning.

Our code, pre-trained models and additional results are available on the project webpage [1].

## 2 Related Work

**Vision-based robotic manipulation.** While earlier methods for solving robotics tasks such as visual servoing [21, 22] were designed manually, the need to cope with large variations of objects and environments led to the emergence of learning-based neural approaches [23, 24, 25, 26]. Deep neural

---

[1]We follow definitions in RLBench [19] for tasks and variations. A task can be composed of multiple variations that share the same skills but differ in objects, attributes or order as shown in Figure 1 (right).

networks [27, 28] have achieved impressive results in manipulation for single tasks [29], and recently led to more challenging setups such as multi-task learning [30, 31, 32, 33]. Different multi-task approaches are explored by discovering which tasks should be trained together [15, 34], determining shared features across tasks [35, 36], meta-learning [37, 38, 39], goal-conditioned learning [40, 41], or inverse reinforcement learning [42]. These approaches can be generally split in two categories according to the training algorithm: reinforcement learning (RL) methods [43, 44, 45, 46] which learn policies from rewards provided by environments and behavioral cloning methods [47, 48, 49] that learn from demonstrations using supervised learning. Demonstrations can be obtained from humans [50], robots [23, 51] or play interactions [49]. The emergence of robotic simulators, such as Gym [52], manipulaTHOR [53], dm_control [54], Sapien [55], CausalWorld [56], and RL-Bench [19], also greatly accelerated the development of manipulation methods. In this work, we use behavioral cloning to train policies given scripted demonstrations from RLBench [19] which covers many challenging manipulation tasks.

**Instruction-driven vision-based robotic manipulation** has received growing attention for manipulations in 2D planar [57, 58] or recent 3D environments [8, 59, 60], and has been transferred to the real world [9, 10]. As grounding the language in visual scenes is important, existing works have focused on challenges in object grounding, such as localizing objects based on referring expressions [61, 62, 63] and grounding spatial relationships [7, 64, 65]. Since language describes high-level actions, several works [60, 66, 67] consider a hierarchical approach to decompose a task into sub-goals. Because natural language is rich and diverse, while training resources are limited, further works learn from collected offline data with instructions [10, 17] or leverage pre-trained vision-language models [4, 5] for action prediction [9, 68]. To further improve the precision of manipulation skills, Mees *et al.* [8] align instructions with multiple cameras by fusing input images with known camera parameters. Most of these works [6, 8, 9, 10] are stateless, since they only employ current observations to predict next actions. Instead, our work proposes to jointly model language instructions, history, and multi-view observations.

**Transformers** [69] have led to significant gains in natural language processing [2], computer vision [70] and related fields [4, 5, 71]. They have also been used in the context of supervised reinforcement learning, such as Decision Transformer [72] or Trajectory Transformer [73]. Recent works in Vision-and-Language Navigation (VLN) [74, 75, 76] further demonstrate that the Transformer allows to better leverage previous observations to improve multi-modal action prediction. Transformers are also used to build a multi-modal, multi-task, multi-embodiment generalist agent, GATO [77]. Inspired by the success of transformers, we explore the transformer architecture for instruction-driven and history-aware robotic manipulation.

# 3  Problem Definition

Our goal is to train a policy $\pi\left(a_{t+1}|\{x_l\}_{l=1}^n, \{o_i\}_{i=1}^t, \{a_i\}_{i=1}^t\right)$ for robotic manipulation conditioned on a natural language instruction $\{x_l\}_{l=1}^n$, visual observations $\{o_i\}_{i=1}^t$, and previous actions $\{a_i\}_{i=1}^t$ where $n$ is the number of words in the instruction and $t$ is the current step. For robotic control, we use macro steps [12] – key turning points in action trajectories where the gripper changes its state (open/close) or velocities of joints are close to zero. We employ an inverse-kinematics based controller to find a trajectory between macro-steps. In this way, the sequence length of an episode is significantly reduced from hundreds of small steps to typically less than 10 macro steps.

The **observation** $o_t$ at step $t$ consists of RGB images $I_t$ and point clouds $P_t$ aligned with the RGB images. $I_t$ is composed of $\{I_t^k\}_{k=1}^K$ RGB images from $K$ cameras, with each $I_t^k$ being of size $H \times W \times 3$ (height, width, 3 channels). Following [15], we use $K = 3$ with cameras on the wrist, left shoulder and right shoulder of the agent, and $H = W = 128$. Similarly, $P_t$ represents point clouds $\{P_t^k\}_{k=1}^K$ from $K = 3$ cameras. A point cloud $P_t^k \in \mathbb{R}^{H \times W \times 3}$ is obtained by projecting a single channel depth image $H \times W$ from the $k$-th camera in world coordinates using known camera intrinsics and extrinsics. Each point in $P_t^k$ has thus 3D coordinates and is aligned with a pixel in $I_t^k$.

The **action space** $a_t$ consists of the gripper pose and its state following the standard setup in RL-Bench [12]. The gripper pose is composed of the Cartesian coordinates $p_t = (x_t, y_t, z_t)$ and its rotation described by a quaternion $q_t = (q_t^0, q_t^1, q_t^2, q_t^3)$ relative to the base frame. The gripper's state $c_t$ is boolean and indicates whether the gripper is open or closed. An object is grasped when it is located in between the gripper's two fingers and the gripper is closing its grasp. The execution of an action is achieved by a motion planner in RLBench.

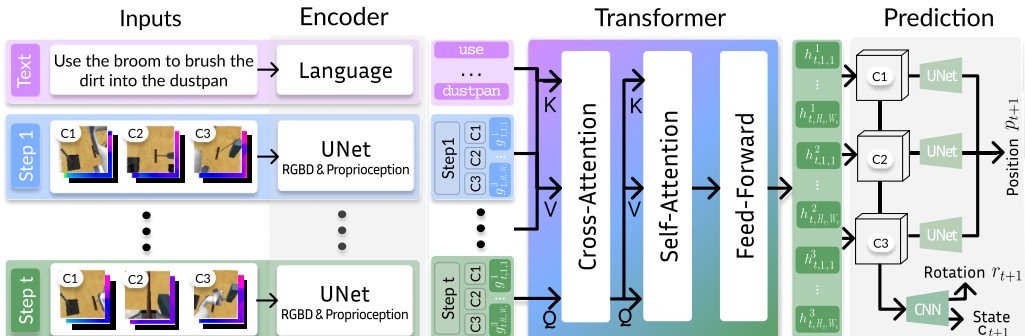

**Figure 2:** Hiveformer jointly models instructions, views from multiple cameras, and past actions and observations with a multimodal transformer for robotic manipulation.

## 4 Our Model: Hiveformer

We propose a unified architecture for robotic tasks called Hiveformer (**H**istory-aware **i**nstruction-conditioned multi-**vie**w trans**former**), see Figure 2 for an overview. It consists of three modules: feature encoding, multimodal transformer and action prediction. The feature encoding module (Sec. 4.1) generates token embeddings for instructions $\{x_l\}_{l=1}^n$, visual observations $\{o_i\}_{i=1}^t$ and previous actions $\{a_i\}_{i=1}^t$. Then, the multimodal transformer (Sec. 4.2) learns relationships between the instruction, current multi-camera observations and history. Finally, the action prediction module (Sec. 4.3) utilizes a convolutional network (CNN) to predict the next rotation $q_{t+1}$ and gripper state $c_{t+1}$, and adopts a UNet decoder [20] to predict the next position $p_{t+1}$.

### 4.1 Feature Encoding

We encode the instruction, visual observations, and actions as a sequence of tokens.

**Instructions.** We employ a pre-trained language encoder to tokenize and encode the sentence instruction. Specifically, we use the language encoder in the CLIP model [5]. Thanks to its vision-and-language pre-training, it is better at differentiating vision-related semantics such as colors compared to pure language-only pre-trained models like BERT [2], see Table 7 in the supplementary material. We freeze the pre-trained language encoder and use a linear layer on top of it to obtain embeddings $\hat{x}_l \in \mathbb{R}^d$ for each word token:

$$\hat{x}_l = \text{LN}(W_x \tilde{x}_l) + E_T^x, \tag{1}$$

with $\tilde{x}_l$ the $l$-th embedding output by the language encoder, LN layer normalization [78], $W_x$ a projection matrix, and $E_T^x$ a type embedding which differentiates instructions from visual observations.

**Observations and Proprioception.** We encode the RGB image $I_t^k$, point clouds $P_t^k$, and proprioception $A_t^k$ for each camera $k$ separately. $A_t^k \in \{0,1\}^{H \times W}$ is a binary attention map used to encode the position of the gripper $p_t$. It takes value one at the location of the gripper center and zero elsewhere. We concatenate $I_t^k$ and $A_t^k$ in the channel dimension and use a UNet encoder to obtain a feature map $\hat{F}_t^k \in \mathbb{R}^{H^v \times W^v \times d_v}$, where $H^v, W^v, d_v$ are the height, width, and number of channels of the feature map. More details about the CNN architecture are presented in Section A of the supplementary material. Next, we concatenate $\hat{F}_t^k$ with point cloud representations in the channel dimension to indicate the spatial location of each patch in the feature map. To match the size of $P_t^k$ and $\hat{F}_t^k$, we apply mean-pooling to $P_t^k$. The final encoded feature map $F_t^k \in \mathbb{R}^{H^v \times W^v \times (d_v+3)}$ is computed as follows:

$$F_t^k = \left[ \text{CNN}([I_t^k; A_t^k]); \text{MeanPool}(P_t^k) \right]. \tag{2}$$

We use patches $f_{t,h,w}^k \in F_t^k, h \in [1, H^v], w \in [1, W^v]$ as separate visual tokens. We further encode $f_{t,h,w}^k$ using embeddings of the camera id $E_C^k$, of the step id $E_S^t$, and of the patch location $E_L^{h,w}$ as well as an embedding to indicate the visual nature of the tokens $E_T^v$ as follows:

$$g_{t,h,w}^k = \text{LN}(W_f f_{t,h,w}^k) + E_C^k + E_S^t + E_L^{h,w} + E_T^v. \tag{3}$$

The encoded visual tokens of the $k$-th camera at step $t$ are denoted as $G_t^k = \{g_{t,h,w}^k\}_{h=1,w=1}^{H^v, W^v} \in \mathbb{R}^{H^v \times W^v \times d}$. We concatenate the encoded tokens for all cameras as $G_t = (G_t^1, \cdots, G_t^K)$.

## 4.2 Multimodal transformer

Given the encoded tokens at the current macro step $t$, the multimodal transformer aims to obtain a contextualized representation for $G_t$ conditioned on the encoded instruction $\{\hat{x}_l\}_{l=1}^n$ and history $\{G_i\}_{i=1}^{t-1}$. This enables learning relationships among views from multiple cameras, the current observations and instructions, and between the current observations and history for action prediction. We use the transformer's attention mechanism [79] to learn such relationships:

$$\text{Attn}(Q,K,V) = \text{Softmax}\left(\frac{W_Q Q (W_K K)^T}{\sqrt{d}}\right) W_V V, \tag{4}$$

where $W_Q, W_K, W_V$ are learnable parameters. Unlike previous work [75] that uses self-attention layers to capture all relationships, we employ different attention layers to capture different types of relationships, in order to reinforce the importance of the context. First, we use a cross-attention layer to learn the inter-modal relationships between $G_t$ and its conditioned contexts $C_t$ consisting of tokens in the instruction $\{\hat{x}_l\}_{l=1}^n$ and history $\{G_i\}_{i=1}^{t-1}$, which is:

$$\tilde{G}_t = \text{CA}(G_t, C_t) = \text{Attn}(G_t, C_t, C_t). \tag{5}$$

Then we learn the intra-modal relationships among patch tokens obtained from the views from multiple cameras through a self-attention layer, *i.e.* $\text{SA}(\tilde{G}_t) = \text{Attn}(\tilde{G}_t, \tilde{G}_t, \tilde{G}_t)$. Finally, a feed-forward network consisting of two linear layers $W_1$ and $W_2$ is applied as follows:

$$\hat{G}_t = \text{LN}\left(W_2 \, \text{GeLU}\left(W_1 \, \text{SA}(\tilde{G}_t)\right)\right). \tag{6}$$

## 4.3 Action Prediction

We concatenate the output embeddings of the transformer $\hat{G}_t$ in Eq (6) and the original encoded visual representations $\hat{F}_t$ in Sec. 4.1 in the channel dimension and reshape the flattened sequence into a feature map $H_t \in \mathbb{R}^{K \times H^v \times W^v \times (d+d_v)}$ to predict the next action $a_{t+1} = [p_{t+1}; q_{t+1}; c_{t+1}]$. As some RLBench tasks require accurate fine-grained positioning, different from the rotation $q_{t+1}$ and gripper state $c_{t+1}$, the position $p_{t+1}$ is predicted through a separate module that uses point clouds $P_t$.

**Rotation and gripper's state.** We transform $H_t$ into $\mathbb{R}^{H^v \times W^v \times K(d+d_v)}$ and feed it into a CNN decoder, described in Section A of the supplementary material. We then apply average pooling across spatial dimensions and employ a linear layer to regress a 5-dimension vector $[q_{t+1}; c_{t+1}]$.

**Position.** The prediction of the gripper position $p_{t+1}$ is decomposed into an expected point on point clouds $p_{t+1}^e$ and an offset $p_{t+1}^o$, *i.e.* $p_{t+1} = p_{t+1}^e + p_{t+1}^o$. The offset allows us to predict a virtual point outside the convex hull of the point cloud, *e.g.* when a robotic arm reaches first above the object and then touch the object. For each camera $k$, a CNN with an upsampling layer predicts an attention map $B_t^k \in \mathbb{R}^{H \times W}$ over the point clouds $P_t^k$. Each value $B_{t,h,w}^k \in B_t^k$ corresponds to the probability of reaching the point $P_{t,h,w}^k \in P_t^k$. Therefore, we compute $p_{t+1}^e$ as the expected position over all cameras:

$$p_{t+1}^e = \sum_{k,h,w} \left(B_{t,h,w}^k \cdot P_{t,h,w}^k\right). \tag{7}$$

The offset $p_{t+1}^o$ is computed from the instruction and the current step id. Let $E_O \in \mathbb{R}^{N_\tau \times T \times 3}$ be a learnable embedding, where $N_\tau$ is the number of tasks and $T$ is the maximum length of episodes. We predict the task id from the instruction: $\text{Pr}(m) = \text{Softmax}\left(W_m \frac{1}{n}\sum_{l=1}^n \tilde{x}_l\right)$, where $\text{Pr}(m) \in [0,1]^{N_\tau}$, and we obtain the offset as: $p_{t+1}^o = \sum_m \text{Pr}(m) \cdot E_O(m, t, :)$.

## 4.4 Training and Inference

**Losses.** We use behavioral cloning to train the models. In RLBench, we generate D, a collection of $N$ successful demonstrations for each task. Each demonstration $\delta \in$ D is composed of a sequence of (maximum) $T$ macro-steps with observations $\{o_i^\delta\}_{i=1}^T$, actions $\{a_i^*\}_{i=1}^T$, task $m^*$ and instruction $\{x_l\}_{l=1}^n$. We minimize a loss function $\mathscr{L}$ over a batch of demonstrations B $= \{\delta_j\}_{j=1}^{|B|} \subset$ D. The loss function is the sum of two losses: a mean-square error (MSE) on the gripper's action and a cross-entropy (CE) over the task classification:

$$\mathscr{L} = \frac{1}{|B|} \sum_{\delta \in \text{B}} \left[\sum_{t \leq T} \text{MSE}(a_t, a_t^*) + \text{CE}(\text{Pr}(m), m^*)\right]. \tag{8}$$

**Table 1:** Success rate on the single-task setting. We report mean and variance for unseen episodes.

| | Inputs | | | Transformer | | | Training | SR |
|---|---|---|---|---|---|---|---|---|
| | Visual Tokens | Point Clouds | Gripper Position | Multi-View | History | Attn | Mask Obs | |
| R1 | × | × | × | × | × | × | × | 72.9±4.1 |
| R2 | Channel | × | × | ✓ | × | Self | × | 73.1±4.5 |
| R3 | Channel | ✓ | × | ✓ | × | Self | × | 77.1±5.8 |
| R4 | Channel | ✓ | ✓ | ✓ | × | Self | × | 78.1±5.8 |
| R5 | Channel | ✓ | ✓ | ✓ | ✓ | Self | × | 81.8±5.2 |
| R6 | Channel | ✓ | ✓ | ✓ | ✓ | Self | ✓ | 82.3±5.3 |
| R7 | Patch | ✓ | ✓ | ✓ | ✓ | Self | ✓ | 84.4±6.4 |
| R8 | Patch | ✓ | ✓ | ✓ | ✓ | Cross | ✓ | 88.4±4.9 |

**Masking current observation.** To ensure that the model uses past information $\{o_i\}_{i=1}^{t-1}, \{a_i\}_{i=1}^{t-1}$ instead of only relying on the current observation $o_t$, we randomly mask the current observation with a probability of 0.1. The masking zeros out randomly selected patch features in the current observation. Therefore, even if the unmasked current observations contain sufficient information, the model still requires to complete the masked observations from the history for action prediction.

## 5 Experiments

In this section we present experiments on RLBench [19] tasks to demonstrate the effectiveness of our Hiveformer model in three settings: single-task, multi-task, and multi-variation. In the single-task setup, a separate model is trained and tested for each task with no variations of the task. Multi-task refers to a setting where one model is trained for multiple tasks (but each task has a unique variation). In the multi-variation case we train a single model to solve multiple variations of a single task and test it on new variations of the task unseen during training.

### 5.1 Experimental Setup

**Dataset setups.** RLBench [19] is a benchmark of robotic tasks. To compare our method with previous work [15], we use the same 10 tasks with 100 demonstrations for training unless stated otherwise. We further evaluate our model on 74 tasks for which RLBench provides successful demonstrations. We manually group these tasks into 9 categories according to their challenges. More details on the split of the tasks are given in Section B of the Supplementary material. We evaluate models by measuring the per task success rate for 500 unseen episodes.

**Implementation details.** We use the Adam optimizer with a learning rate of $5 \times 10^{-5}$. Each batch consists of 32 demonstrations. Models were trained for 100,000 iterations. We apply data augmentation in training including jitter over RGB images $I_t^k$, and a random crop of $I_t^k$, $P_t^k$, and $A_t^k$ while keeping them aligned. Models are trained on one NVIDIA Tesla V100 SXM2 GPU using a Singularity container with headless rendering. Auto-$\lambda$ [15] uses a UNet network and applies late fusion to predictions from multiple views.

### 5.2 Ablations

To demonstrate the effectiveness of the proposed model architecture, we ablate the impact of its components in Table 1. The model in R1 (row 1) is a UNet architecture similar to Auto-$\lambda$ [15] except that it is conditioned on instructions rather than task ids. This baseline only uses visual observations at the current step and already achieves promising results with a success rate of 73.2%. On top of R1's architecture, a multimodal transformer with self-attention is added in R2 to improve the modeling of multi-view images. Visual tokens $\{G_t^i\}_{i=1}^K$ are different channels in the feature map instead of spatial patches used in our final model. In R3 and R4, we further add point clouds $P_t$ and gripper position $A_t$ in the feature encoding, which leads to 3% improvement in total. The impact of history, *i.e.* the use of observations from previous steps, $(\{G_j^i\}_{i=1,j=1}^{K,t-1})$ is studied in R5 and R6. The history information brings 4.5% absolute gains and the masking of observations during training further improves the performance by 0.5%. In R7, we replace the tokenization of feature maps from channels $g_{t,c}^k$ to patches $g_{t,h,w}^k$, and obtain another 2.2% gain. This improvement can be attributed

**Table 2:** Comparison with state-of-the-art methods on 10 tasks. We report success rate (%).

| | Pick & Lift | Pick-Up Cup | Push Button | Put Knife | Put Money | Reach Target | Slide Block | Stack Wine | Take Money | Take Umbrella | Avg. |
|---|---|---|---|---|---|---|---|---|---|---|---|
| *Single-task learning* | | | | | | | | | | | |
| ARM [12] | 70 | 80 | - | - | - | 100 | - | 70 | - | 70 | - |
| Auto-$\lambda$ [15] | 82 | 72 | 95 | 36 | 31 | 100 | 36 | 23 | 38 | 37 | 55.0 |
| Ours | 92.2 | 77.1 | 99.6 | 69.7 | 96.2 | 100.0 | 95.4 | 81.9 | 82.1 | 90.1 | 88.4 |
| *Multi-task learning* | | | | | | | | | | | |
| Auto-$\lambda$ [15] | 87 | 78 | 95 | 31 | 62 | 100 | 77 | 19 | 64 | 80 | 69.3 |
| Ours (w/o inst) | 83.8 | 13.9 | 97.0 | 41.9 | 54.3 | 98.9 | 36.2 | 68.5 | 74.1 | 73.0 | 64.2 |
| Ours | 88.9 | 92.9 | 100.0 | 75.3 | 58.2 | 100.0 | 78.7 | 71.2 | 79.1 | 89.2 | 83.3 |

**Table 3:** Comparison with the state of the art on 74 RLBench tasks grouped into 9 categories. We report success rate (%) for the single-task setting. *The performance of Auto-$\lambda$ is obtained by running their code.

| | Planning | Tools | Long Term | Rot. Invar. | Motion Planning | Screw | Multi Modal | Precision | Visual Occlusion | Avg |
|---|---|---|---|---|---|---|---|---|---|---|
| Num. of tasks | 9 | 11 | 4 | 7 | 9 | 4 | 5 | 11 | 14 | 74 |
| Auto-$\lambda$ [15]* | 58.9 | 20.0 | 2.3 | 73.1 | 66.7 | 48.2 | 47.6 | 34.6 | 40.6 | 44.0 |
| Ours (w/o hist) | 78.9 | 46.7 | 10.0 | 84.6 | 73.3 | 72.6 | 60.0 | 63.8 | 57.9 | 60.9 |
| Ours (one view) | 57.7 | 23.2 | 12.3 | 57.8 | 63.2 | 35.6 | 40.7 | 33.7 | 37.1 | 40.1 |
| Ours | 81.6 | 53.0 | 16.9 | 84.2 | 72.7 | 80.9 | 67.1 | 64.7 | 60.2 | 65.4 |

to patch tokens that help encode fine-grained spatial information. Finally, we use cross-attention instead of self-attention (Eq. 5) to condition on the instruction and history context. It further boosts the performance with a 3.8% gain.

## 5.3 Comparison with State of the Art

**Single-task evaluation.** The upper block in Table 2 presents results of different models on 10 single tasks in RLBench. We compare our model with ARM [12] and Auto-$\lambda$ [15], two state-of-the-art methods on RLBench and observe a consistent improvement for all tasks.

**Extending tasks in a single-task evaluation setup.** In Table 3, we further compare Auto-$\lambda$ and Hiveformer's variants across 74 RLBench tasks grouped into 9 categories. The variant without history removes the history tokens in Hiveformer, while the variant with one view only uses one camera at each step (we take the best among the 3 cameras for each task). The full Hiveformer achieves consistently better performance compared to Auto-$\lambda$ [15] on all types of tasks. Among them, the Long-term, Tools and Planning task groups assess the use of history, where our model brings improves significantly over the variant without history. Compared to the one view variant, our full model performs significantly better on tasks requiring fine-grained control or with large occlusions such as Screw, Precision and Visual Occlusion categories. Yet, our method performs relatively poorly for Long-term tasks with more than 10 steps, such as "take shoes out of box". As Long-term tasks have an average number of steps 2-4 times higher than others, they are more prone to distribution shift issues and accumulated errors. Hierarchical modeling or better training algorithms such as reinforcement learning and dagger [80] could be helpful, but are left as future work.

**Multi-task evaluation.** The lower half in Table 2 shows the results in a multi-task setting. Notably, Auto-$\lambda$ uses a training algorithm which dynamically adjusts the weights of different tasks, while our model simply treats all tasks with equal weights. Nevertheless, our model outperforms Auto-$\lambda$ by 14%, demonstrating the improvements due to our architecture. We further compare our model with a variant without instructions in the input sequence (since $p_{t+1}^o$ is predicted from instructions, we modify the model such as it is predicted from $H_t^k$). The results show that instructions are important in the multi-task setting. Moreover, the performance of our *single* model trained for all tasks is only slightly worse than the performance of individual models for each task.

**Table 4:** Success rate (%) in the multi-variation setting for seen or unseen variations and synthetic or human-written instructions.

| # Demos | Push Buttons | | | Tower | | |
| Per | Seen | Unseen | | Seen | Unseen | |
| Variation | Synthetic | Synthetic | Human | Synthetic | Synthetic | Human |
|---|---|---|---|---|---|---|
| 10 | 96.8 | 73.1 | 65.1 | 71.7 | 50.1 | 19.4 |
| 50 | 99.6 | 83.3 | 70.6 | 74.1 | 52.3 | 20.7 |
| 100 | 100 | 86.4 | 74.0 | 76.2 | 56.4 | 24.2 |

**Generalization to multi-variations.** Table 4 shows results of Hiveformer trained on different variations of the two tasks *Tower* and *Push Buttons*. The *Tower* (resp. *Push Buttons*) task requires the robot to sequentially stack colored cubes (resp. push colored buttons) using the order specified in the instruction, see Figure 1 (right).

We use 100 variations in training and test models for both the 100 seen variations and 100 unseen variations. In this setting, instructions are necessary to generalize to unseen variations (*e.g.* it is impossible to distinguish the order of pushing buttons red-green-blue vs. blue-red-green by only looking at the scene). We compare the models trained with different numbers of demonstrations per variation. Even in the most challenging case where only 10 demonstrations are available per variation, Hiveformer achieves a success rate of 71.1% for the *push buttons* task and 49.8% for the *tower* task in unseen variations. Furthermore, besides tests on synthetic instructions (Synt), we also test the generalization to real instructions. Despite being only trained on synthetic instructions with limited vocabulary and diversity, our model performs well on instructions generated by humans (Real). Finetuning Hiveformer on human instructions [75] is expected to result in further improvements. Details of human-generated instructions are presented in Section C of the Supplementary material.

### 5.4 Real-robot Experiments

**Setup.** We conduct real-robot experiments for the *push buttons* task on a 6-DoF UR5 robotic arm equipped with a 2-finger Robotiq RG2 gripper and two cameras on each side of the scene. As there exists a large difference between simulated and real environments, we finetune the simulator-trained policy on real-robot demonstrations. We use 10 variations of the task and 10 real-robot demonstrations per variation. More details are presented in Section E of the Supplementary material.

**Results.** We report success rates for real-robot experiments in Table 5 on the "push buttons" task using synthetic instructions. The models are tested on 10 seen and 10 unseen variations. We compare two models: one trained from scratch using real-robot demonstrations; and the other pretrained on RLBench and then finetuned using real-robot demonstrations. As shown in Table 5, the pretraining significantly improves the performance especially for unseen variations. The model without pretraining is prone to overfitting on seen variations. Although the domain gap between the real robot and RLBench environments is large, our model benefits from pretraining in the simulator. More analysis and examples are presented in Section E of the Supplementary material.

**Table 5:** Success rate of push buttons task on real robots.

| Pretrain | Seen Vars | Unseen Vars |
|---|---|---|
| - | 86.7 | 13.3 |
| ✓ | 92.2 | 85.7 |

## 6 Conclusion

We introduced Hiveformer, a multimodal transformer that jointly models instructions, views from multiple cameras, and history for instruction-driven robotics manipulation. We evaluated the model on RLBench in three settings: single-task learning, multi-task learning, and multi-variation generalization and demonstrated its effectiveness outperforming state of the arts. We deployed our model on a real robot which is able to generalize to unseen variations and human-written instructions.

**Limitations.** The computational cost quadratically increases with the input sequence length due to the transformer. Furthermore, our model is trained with behavioral cloning, which may suffer from exposure bias. Future works could improve the efficiency for long-term tasks with hierarchical models and also incorporate reinforcement learning. Moreover, our model is trained on only synthetic instructions and performs worse on the human-written instructions. Training on human-written automatically generated instructions could help improve the performance.

**Acknowledgments**

This work was granted access to the HPC resources of IDRIS under the allocation 101002 made by GENCI. This work is funded in part by the French government under management of Agence Nationale de la Recherche as part of the "Investissements d'avenir" program, reference ANR19-P3IA-0001 (PRAIRIE 3IA Institute), the ANR project VideoPredict (ANR-21-FAI1-0002-01) and by Louis Vuitton ENS Chair on Artificial Intelligence.

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
