# OpenReview forum: "Instruction-driven history-aware policies for robotic manipulations"
_robot-learning.org/CoRL/2022/Conference — CoRL 2022 Oral_

### Official Review · Reviewer_R4uU · 2022-07-28

**Originality:** Good
**Technical Quality:** Good
**Clarity Of Presentation:** Good
**Impact:** 3

**Recommendation:**

Weak Accept: I recommend accepting the paper, but will not argue for my recommendation if the majority of other reviewers have a different opinion.

**Summary:**

The work introduces a transformer-based robot manipulation method to execute language instructions. The policy input contains the history of actions and observations, which enables the execution of sequential tasks. The method also integrates a multi-camera view, in order to deal with the need for precise control and issues like occlusions.
The method was evaluated in simulated RLBench in single-task, multi-task, and multi-variation setup. The method was also demonstrated on a real robot (the results are only in the supplementary material).

**Issues:**

- Three setups for the RL bench experiments (single-task, multi-task, and multi-variation) should be defined or at least explained.
- Please provide detail on how macro steps (line 115) are used.
- Following the limited (in my opinion) evaluation of the language component, I'd like to learn answers to some of the following questions:
    - how well the method behaves, if we use a large-language model embedding of the instructions
    - lines 144-146 claim that CLIP embedding is better than BERT. Do you have any evidence to back this claim?
    - is the $E_T^x$ in Eqn (1) useful/required?
    - Does the model use task similarity expressed in language in the multi-task setup? How would the performance change, if we use a one-hot encoded representation for tasks, instead of the language input?
- Could you please provide examples of the human-written instructions? The description in the supplementary does not give me an insight, into how diverse they were (especially for the push-button task). It would be great if the full set of human-written instructions is provided in the supplementary material.
- I think that the limitation section should include a discussion on how well the method handles *natural* language / human-written instructions (as claimed in the abstract and conclusion line 309). Most of the experiments are using static descriptions of tasks. Using human-written instructions can reduce the performance up to 3x (from 77.4 to 21.3 in the tower task), which I'm not sure if it justifies the "generalizes well" claim (line 309).
    - Alternatively, support evidence that the model can actually handle language instructions. I could imagine an experiment in the multi-task setup, with synonyms for the task description (e.g. "Lamp on" -> "Turn the light on"). A similarly interesting experiment would be to show a transfer from a model trained on 73 tasks to a different one, not seen in the training (e.g. one of the "put/close/open" tasks, which would have similar concepts in the training set).

**Quality Of The Limitations Section:**

Additional details required

**Reviewer Expertise:**

3: The reviewer is fairly confident that the evaluation is correct

**Robotics Focus:**

Sufficient demonstration on hardware

**Strengths And Weaknesses:**

Strengths:
- good empirical results on RLBench, compared to ARM and Auto-$\lambda$
- good results on the real robot. In particular, it shows the significant benefit of pretaining in simulation (and transfer from 3 to 2 cameras)
- a thorough analysis (Section 5.2) of the importance of different components of the perception and manipulation processing (not the language though).

Weaknesses:
- a significant part of the RLBench experiments is in the single-task setup, which if I understand it correctly, does not need the instructions at all (a separate model is trained for each task). Thus we get relatively little insight on the "instruction-driven" part. We can learn from Table 3 that it does help, but we don't know, how it compares to other ways of processing the language input. The "average over word tokens" presented in Table 4 seems to be very crude.
- the description of the architecture (Sect. 4.1, 4.2, Fig 2) seems to be pretty difficult to comprehend.
- The source code does not come with instructions, on how to run it.
- The source code contains the name of the author in the file `pyproject.toml`. It should be anonymized.

**Summary Of Recommendation:**

The presented method has many strenghts. It is a complete solution for executing multi-task, potentially multi-step instructions. The additional benefit is that it can leverage multiple cameras and a history of actions and observations. The method was also demonstrated to work on a robot (in time for the supplementary material deadline), which is very important for the relevance to the conference. Finally, the RLBench results seem strong.

On the other hand, the novelty of the method is somehow limited. Moreover, the claims about handling natural-language instructions do not seem to be sufficiently supported by the experiments.

---

> ### Author Response · Authors · 2022-08-27
> **Answer to reviewer #3 (R4uU) [1/2]**
>
> We thank the reviewer for the positive and constructive comments.
>
> **A significant part of the RLBench experiments is in the single-task setup, which does not need the instructions. Thus we get relatively little insight on the "instruction-driven" part. We can learn from Table 3 that it does help, but we don't know how it compares to other ways of processing the language input. The "average over word tokens" presented in Table 4 seems to be very crude.**
>
> Since existing works on RLBench [11, 84, R1] mainly focus on the single-task setup, we perform most of the ablation studies on the single-task setting.
>
> We clarified the single-task, multi-task and multi-variation setting in lines 212-216. The instructions are used in the multi-task and multi-variation settings. We demonstrated that the instructions play an important role in the two settings (Tables 2 & 4 in the revised paper).
>
> We agree that the instruction encoding method (“average over word tokens”) is a very simple baseline. However, we have included it to compare to previous works such as CLIPort [8]. The results confirm that our proposed method is more effective.
>
> [R1] Chen, Bryan, et al. "Robust Policies via Mid-Level Visual Representations: An Experimental Study in Manipulation and Navigation." Conference on Robot Learning. PMLR, 2021.
>
>
> **The source code does not come with instructions, on how to run it.  Anonymize.**
>
> We updated the code accordingly in the revised supplementary material.
>
> **Three setups for the RL bench experiments (single-task, multi-task, and multi-variation) should be defined or at least explained.**
>
> The single-task setup means that the training and testing set contains a single task with a single variation of this task. The multi-task refers to a setting where a single model is trained over multiple tasks (but each model has a unique variation). The multi-variation case is when a single model is trained with multiple variations of a single task. Thanks for pointing this out, we have now clarified this in the revised version (lines 212-215).
>
> **Please provide detail on how macro steps (line 115) are used.**
>
> Our model predicts the position of the gripper for each macro step. An inverse-kinematics based controller is employed to find a trajectory between macro steps. This setup follows the exact same approach of previous work (ARM and Auto-$\lambda$). We further clarified it in the revised version (lines 116-117).
>
> **How well the model behaves with BERT embeddings? Without $E^x_T$? With a one-hot encoded representation instead of CLIP embeddings? Does the model use task similarity expressed in language in the multi-task setup?**
>
> We run additional ablation studies in the multi-variation setup for the push buttons task. We found that BERT embeddings reduce the performance by 43.0% on synthetic instructions and 52.4% on human-written instructions. This can be explained by the fact that the BERT embeddings pretrained on pure texts do not differentiate well different colors. We found that removing $E^x_T$ decreases the performance only by 3.1%, but removing both $E^x_T$ and $E^v_T$ decreases the performance by 21.1%.
> Moreover, replacing the instructions with a one-hot encoding of the variation index increases the performance in the seen variations (by 19.4 % on the tower task), but it prevents the model from generalizing to unseen variations.
> We added these ablations in the supplementary material (Table 7) of our revised version.
>
> **Could you please provide examples of the human-written instructions?**
>
> The full set of instructions has been added in the supplementary material in the annotations.json file. We also provided a few examples in the main paper of the revised version (line 299-302). They contain unseen verbs (e.g. “Tap on the green button, then the grey button and
> end up pressing the pink button”), unseen formulations (e.g. “Press the green, cyan and pink buttons
> in that order”), longer sentences (e.g. “Press the while button and then you go to green button and
> press it and finally press the black button”) or unseen color references (e.g. “Press the darker blue
> button, then the gray one and finally the lighter blue button.”).

---

> > ### Author Response · Authors · 2022-08-27
> > **Answer to reviewer #3 (R4uU) [2/2]**
> >
> > **The limitation section should include a discussion on how well the method handles natural language / human-written instructions (as claimed in the abstract and conclusion line 309).**
> >
> > Human-written instructions are more varied than the synthetic ones. They contain unseen verbs (e.g. “Tap on the green button, then the grey button and end up pressing the pink button”), unseen formulations (e.g. “Press the green, cyan and pink buttons in that order”), longer sentences (e.g. “Press the white button and then you go to green button and press it and finally press the black button”) or unseen color references (e.g. “Press the darker blue button, then the gray one and finally the lighter blue button.”).
> > Therefore, our model trained on only synthetic instructions performs worse on the human-written instructions (86.3% -> 74.2% on the push buttons task and 56.2% -> 24.1% on the tower task). We could annotate or automatically generate more natural instructions to train/finetune our model with human-written instruction in the future work, which has shown to be beneficial in prior work [77]. We have added the discussion in the limitation section in the revised version.

---

### Official Review · Reviewer_yM9G · 2022-08-01

**Originality:** Good
**Technical Quality:** Excellent
**Clarity Of Presentation:** Excellent
**Impact:** 3

**Recommendation:**

Strong Accept: I recommend accepting the paper and will argue for my recommendation even if other reviewers hold a different opinion.

**Summary:**

This paper presents Hiveformer (History-aware instruction-condition multi-view transformer), which enables a robot to learn to perform novel manipulation tasks based upon natural language instructions and demonstrations for a set of training tasks. The approach uses Behavior Cloning to train the model, and the actions consist of gripper poses. A key aspect of the work is encoding multiple views of the scene and a history of information. Results in RLBench show that the approach outerpforms ARM [11] and Auto-\lambda [14] as well as ablations of Hiveformer. A video is included and depicts a robot demonstration.

**Issues:**

Please address the notes on limitations, missing prior work, some minor issues of clarity, and the baseline of Silva et al. (2022).

**Quality Of The Limitations Section:**

Additional details required

**Reviewer Expertise:**

4: The reviewer is confident but not absolutely certain that the evaluation is correct

**Robotics Focus:**

Sufficient demonstration on hardware

**Strengths And Weaknesses:**

Strengths:
-The figures are clear and useful.
-The right amount of space is allocated to the various parts of the paper.
-The presentation of the algorithm is intelligible.
-Code is provided.
-The method shows positive results.

Weaknesses:
-It seems from Figure 1 that the paper addresses multi-step tasks; however, it isn't quite clear what examples of this task are and where the results are. I would guess those results are in Table 2 under "Long Term." However, it would be helpful to provide more detail.
-Some of the success rates are quite low (e.g., 16.9%). It would be helpful to discuss what it would take to reach > 70% on call tasks. On that note, it would be helpful if the table captions confirmed that what is show are the success rates (as opposed to rewards, etc.)
-It would have been helpful to show an ablation study evaluating the quality of the algorithm's performance as a function of the number of demonstrations available. 100 demonstrations per task could be too much for a human demonstrator. Further, how much variation existed among the demonstrations, and how were these demonstrations collected?
-Some tables (etc.) go over the margins for the paper and should be adjusted accordingly.
-This paper might have missed related work in language-guided robot manipulation. In particular, please see the references below. The first two address only RL, whereas the last one addresses imitation learning-based manipulation with language. As such, it would have been helpful to benchmark against the Silva et al. work.

Sodhani, S., Zhang, A. and Pineau, J., 2021, July. Multi-task reinforcement learning with context-based representations. In International Conference on Machine Learning (pp. 9767-9779). PMLR.

Stengel-Eskin, E., Hundt, A., He, Z., Murali, A., Gopalan, N., Gombolay, M. and Hager, G., 2022, January. Guiding Multi-Step Rearrangement Tasks with Natural Language Instructions. In Conference on Robot Learning (pp. 1486-1501). PMLR.

Silva, A., Moorman, N., Silva, W., Zaidi, Z., Gopalan, N. and Gombolay, M., 2021. LanCon-Learn: Learning With Language to Enable Generalization in Multi-Task Manipulation. IEEE Robotics and Automation Letters, 7(2), pp.1635-1642.

-CLIP, as noted by Abeba Birhane et al. (2021), is problematic. The authors leverage CLIP without discussing the potential harm that can come from its adoption. A recent paper exploring negative impacts of CLIP in robots was published by Hundt et al. (FAccT'22), but this was published after this paper's CoRL submission. It would be helpful if the paper addressed this issue in the limitations.

Birhane, A., Prabhu, V.U. and Kahembwe, E., 2021. Multimodal datasets: misogyny, pornography, and malignant stereotypes. arXiv preprint arXiv:2110.01963.

Hundt, A., Agnew, W., Zeng, V., Kacianka, S. and Gombolay, M., 2022, June. Robots Enact Malignant Stereotypes. In 2022 ACM Conference on Fairness, Accountability, and Transparency (pp. 743-756).

**Summary Of Recommendation:**

This paper is solid. The presentation is clear, the method is well-reasoned, and the results are positive. Ablations are provided, and the method outperforms baselines. It would have been helpful to include the work by Silva et al. (2022) as a direct baseline.

---

> ### Author Response · Authors · 2022-08-27
> **Answer to reviewer #2 (yM9G) [1/2]**
>
> We thank the reviewer for the positive and constructive comments.
>
> **It seems from Figure 1 that the paper addresses multi-step tasks; however, it isn't quite clear what examples of this task are and where the results are. I would guess those results are in Table 3 under "Long Term".**
>
> In this work, we define a step as the key turning point in action trajectories of an agent (L116-117). Given this definition, all tasks except for the reaching task in RLBench contain multiple steps and thus can be referred to as multi-step tasks.
> The “Long Term” task group denotes several tasks in RLBench with more than 10 steps. Examples for “long term” tasks are wipe desk, stack blocks, take shoes out of box, slide cabinet open and place cups that all require complicated motions (L585-586 in the supplementary material).
>
>
> **Some of the success rates are quite low (e.g., 16.9%). It would be helpful to discuss what it would take to reach > 70% on call tasks.**
>
> While providing a significant improvement over the SOTA approach Auto-Lambda (+14.6%), our method performs relatively poorly for long term tasks with more than 10 steps. Since the average number of steps in long term tasks is 2-4 times higher than in others, they are more prone to the distribution shift issue and accumulated errors. Approaches such as hierarchical modeling or better training algorithms such as reinforcement learning and dagger could be helpful to address the long-term tasks. We added the discussion in Section 5.3 of the revised version (line 268-272).
>
> **On that note, it would be helpful if the table captions confirmed that what is shown are the success rates (as opposed to rewards, etc.)**
>
> All tables show success rates. We updated the captions accordingly in the revised version.
>
> **It would have been helpful to show an ablation study evaluating the quality of the algorithm's performance as a function of the number of demonstrations available. 100 demonstrations per task could be too much for a human demonstrator.**
>
> We use 100 demonstrations to fairly compare to Auto Lambda—they always use 100 demonstrations (see L219). We evaluate the influence of the number of demonstrations in the multi-variation setting in Table 4. We compared models with 100 variations that are trained with 10, 50 and 100 demonstrations per variation (first column): 50 demonstrations. We observed that 50 demonstrations per variation are sufficient to obtain a satisfying success rate for the push buttons task (the success rate decreased by 3.2% between row 3 and row 2 vs 15.2% between row 3 and row 2). Note that Table 4 provides success rates obtained in simulation. For our real robot experiments, we spent a couple of hours to collect 10 real demonstrations per variation and 10 variations for finetuning a model pretrained with simulated RLBench data (Table 5) to achieve good performance.
>
>
> **How much variation existed among the demonstrations, and how were these demonstrations collected?**
>
> RLBench has built-in scripts to generate ground truth demonstrations. They randomize the initial location of objects over the entire workspace (80 cm x 30 cm). For the real-world demonstrations, we also create random initial configurations and use an expert script to generate ground-truth actions to solve the task in the real world.
>
> **Some tables go over the margins for the paper and should be adjusted accordingly.**
>
> Thanks, we have adjusted them in the revised version.
>
> **This paper might have missed related work in language-guided robot manipulation.**
>
> Thank you for pointing out the references. We have added them to the revised version in the related work.

---

> > ### Author Response · Authors · 2022-08-27
> > **Answer to reviewer #2 (yM9G) [2/2]**
> >
> > **It would have been helpful to benchmark against the Silva et al. work.**
> >
> > LanCon-Learn’s approach takes as input the gripper state and the object state, namely the ground truth pose of each object in the scene, instead of raw visual observations as ours. It encodes textual instructions with GloVe embeddings and a bi-directional LSTM. It predicts the next pose of the gripper based on a modular architecture conditioned on encoded text features.
> > We run experiments on RLBench with the code provided by the authors. Since some RLBench tasks require identifying the colors of an object, we modified their object state to include a RGB reference of each object.
> > Moreover, we complete their method with a history mechanism, where the gripper state is concatenated with the gripper state from the previous step.
> >
> > As presented in Table 6 in the revised supplementary material, we obtained an average success rate of 63.9% with their original method (vs. 72.9% LanCon-Learn with history vs. 88.3% for our approach) in our single-task setting and 47.8% (vs. 57.4% with history vs. 83.8% for our approach) in our multi-task setting.
> > In the multi-variation setting disclosed in Table 7 in the Supplementary Material, the gap appears wider, since GloVe embeddings differentiate poorly colors: for the “push buttons” task on unseen variations and synthetic instructions, the performance reaches only $1.7\%$ (vs. $16.7\%$ with history vs. $86.3\%$ with our approach).

---

### Official Review · Reviewer_G4rm · 2022-08-02

**Originality:** Very Good
**Technical Quality:** Very Good
**Clarity Of Presentation:** Very Good
**Impact:** 3

**Recommendation:**

Weak Accept: I recommend accepting the paper, but will not argue for my recommendation if the majority of other reviewers have a different opinion.

**Summary:**

This paper proposes a model for learning to predict end-effector keypoints to solve manipulation tasks. The model consists of attention mechanisms modeling relationships between text tokens and visual patches, including across time. It also includes a U-Net that generates a spatial probability over gripper locations. The input to the model are RGBD observations from three different views: left shoulder, right shoulder and gripper, and also a tokenized instruction where tokens are encoded with the CLIP encoder.
The paper demonstrates SOTA performance on 10 RLBench manipulation tasks in simulation, includes extensive ablation studies of the different model parts, and also includes an experiment with two simple instruction-conditional tasks. One of the main points made by the paper is that it's important to model the history of observations. The paper provides empirical proof to this claim.

**Issues:**

Questions:
- Line 159: why applying “max-pooling” to point clouds makes sense? Wouldn’t median-pooling (or even average-pooling) make more sense? Wouldn’t max-pooling select the highest coordinate along the x, y, and z axis individually, and take the resulting point to represent the location of the patch? Wouldn’t this representation be highly sensitive to an arbitrary choice of the origin? And e.g. for small objects, wouldn’t it be likely to select far away background points instead of a closer object point to represent the patch location?
- Equation 6: what is LN? The inner layer was represented by explicitly writing down a matrix multiplication. Why does the outer layer use LN and what does it mean?
- Equation 7: If the distribution is multi-modal, then averaging coordinates (e.g. points) may result in completely false predictions in-between the two modes, rather than better predictions on one of the two modes. Is this ever a problem in practice?

Suggestions:
- Line 149 - what type is the token E_{T}^{x}? At this point in reading I have not yet learned about how visual observations are represented or used downstream, so saying “to differentiate instructions from visual observations” doesn’t tell me much.
- Line 151 - maybe call A_{t}^{k} a “binary mask” instead of “attention map”? The word “attention” can have connotations of it being a soft mask, a mask that sums to 1, and a mask computed by a learned model for downstream use in the same or different learned model.
- Line 181 - would be nice to have specific section numbers and links when referring to supplementary material.
- There are multiple prior works on instruction-following (e.g. [A, B, K, L]), that build maps as a means to provide a concise spatial memory to a robot. This is often seen as an advantage over keeping in memory a complete history of observations. Could the paper discuss why keeping history might be advantageous?

Comments:
- Section 4.2: I’m not sure it is fair to call this method a “multi-modal transformer”. The attention mechanism was invented before the Transformer architecture for machine translation, and one important aspect of it is that the attention modules are layered in a deep network. The architecture in this paper consists of a single cross-attention and a single self-attention layer, followed by an MLP. Calling it something like “attention across modalities and time” model would more clearly pinpoint the contribution. (the whole time reading this paper I was trying to understand how it is fundamentally different from all the other transformer architectures, and this section finally got to it).

**Quality Of The Limitations Section:**

Limitations are addressed clearly

**Reviewer Expertise:**

3: The reviewer is fairly confident that the evaluation is correct

**Robotics Focus:**

Highly relevant to robotics but no hardware experiments

**Strengths And Weaknesses:**

Strengths:
- This paper doesn’t just take an off-the-shelf transformer and feed text and visual observations through it, but carefully considers what types of attention are required to model the task, and how it should fit together in an architecture. The result is a well-performing and lightweight model specifically targeted at fixed-base manipulation tasks.
- Technical sections very nicely written, with clear definitions to the different components, indexing that makes sense and helps understand the data types involved, and a clear data flow through the model.
- Impressive performance on 10 RLBench tasks.
- Overall strong experimental setup, with a very good set of ablations to study the effects of the different modeling choices like instruction input, history input, cross- vs self-attention, etc.

Weaknesses:
- In terms of studying instruction-driven manipulation, the sort of instructions and generalization studied is relatively extremely simple and significantly behind prior work. E.g. the paper did not test interesting compositions of attributes or spatial relations [C, D, E, F], nor did they test generalization to unseen objects or relations [G, H], nor tasks requiring multi-step planning [J]. As such, calling it an instruction-driven manipulation in the title might be a disservice to the paper, since this aspect is relatively underlooked. There is a large body of work that does instruction-following including modeling of history [B, I, K]. On the other hand, it is a fairly strong simulated table-top manipulation paper in the multi-task behavior cloning setting. Perhaps focusing on this aspect might better highlight the positive attributes of the paper.

- The method for computing position offsets (Lines 198-201) doesn’t seem like it could handle cases where the same task at the same time-step may require a different offset based on the environment configuration. This reduces the generality of the approach. Furthermore, the assumption of a fixed set of tasks (also present in the loss in Equation 8) cements this as a multi-task learning approach rather than an instruction-driven approach. In instruction-driven work, it's questionable to map to a fixed set of tasks, because the set of tasks is by definition extensible via specification of new instructions.

- No experiments on real robots, so tolerance to noise, or sim2real generalization couldn’t be assessed.

- Some design choices seem somewhat arbitrary and maybe strange (e.g. applying max-pooling to point-cloud patches, classifying instructions into a fixed set of task types).


Review references:
[A] Walter, Matthew R., et al. "Learning semantic maps from natural language descriptions." Robotics: Science and Systems, 2013.
[B] Blukis, Valts, et al. "Learning to Map Natural Language Instructions to Physical Quadcopter Control using Simulated Flight." CoRL. 2019.
[C] Arumugam, Dilip, et al. "Grounding natural language instructions to semantic goal representations for abstraction and generalization." Autonomous Robots 43.2 (2019): 449-468.
[D] Vaezipoor, Pashootan, et al. "Ltl2action: Generalizing ltl instructions for multi-task rl." International Conference on Machine Learning. PMLR, 2021.
[E] Wang, Christopher, et al. "Learning a natural-language to LTL executable semantic parser for grounded robotics." Conference on Robot Learning. PMLR, 2021.
[F] Corona, Rodolfo, et al. "Modular Networks for Compositional Instruction Following." Proceedings of the 2021 Conference of the North American Chapter of the Association for Computational Linguistics: Human Language Technologies. 2021.
[G] Shridhar, Mohit, Lucas Manuelli, and Dieter Fox. "CLIPort: What and Where Pathways for Robotic Manipulation." 5th Annual Conference on Robot Learning. 2021.
[H] Blukis, Valts, Ross Knepper, and Yoav Artzi. "Few-shot Object Grounding and Mapping for Natural Language Robot Instruction Following." Conference on Robot Learning. PMLR, 2021.
[I] Pashevich, Alexander, Cordelia Schmid, and Chen Sun. "Episodic transformer for vision-and-language navigation." Proceedings of the IEEE/CVF International Conference on Computer Vision. 2021.
[J] Paxton, Chris, et al. "Prospection: Interpretable plans from language by predicting the future." 2019 International Conference on Robotics and Automation (ICRA). IEEE, 2019.
[K] Anderson, Peter, et al. "Chasing ghosts: Instruction following as bayesian state tracking." Advances in neural information processing systems 32 (2019).

**Summary Of Recommendation:**

The main strength is superior performance on RLBench tasks compared to prior behavior-cloning work, along with great technical execution, good empirical work. However, it's not clear that the paper proposes anything paradigm-shifting. The method utilizes various techniques from prior work (such as the modelling keypoints, cross-modal attention, history modelling, U-Nets, multimodal transformers). None of these can be claimed as a contribution. However, the paper shows a good way of putting these pieces together in a robotics system.

---

> ### Author Response · Authors · 2022-08-27
> **Answer to reviewer #1 (G4rm) [1/3]**
>
> We thank the reviewer for acknowledging our contributions and for the constructive comments to our work.
>
> **The sort of instructions and generalization studied is relatively simple. The paper did not test interesting compositions of attributes or spatial relations, generalization to unseen objects or relations, or multi-step planning. As such, calling it an instruction-driven manipulation in the title might be a disservice to the paper, since this aspect is relatively underlooked.**
>
> Our work mainly utilizes synthetic instructions in RLBench. Different from previous works [R1, R2, R3] which consider the task of “placing a peach into a red bowl” a different task to “placing an apple into a green bowl”, the RLBench makes a distinction between tasks and variations. For example, “placing an object into a container” is a task and the different instantiation of the “container” are variations of the task. Each variation comes with several synthetic instructions to describe the specific objective. In our work, we studied the generalization to multiple variations. This multi-variation setting is similar to prior works [R3] that study the generalization to multiple tasks with different compositions.
>
> Specifically, the experiments on variations of “pushing buttons” and “tower building” tasks involve new compositions of objects and multi-step planning (Table 4, column Unseen). We also studied the generalization to human-written instructions beyond the synthetic instructions (Table 4, column Real). The human-written instructions contain unseen color references (“lighter blue”, “greenish blue”), unseen composition of colors (such as the variations of the button task), and unknown verb actions and sentence formulations (“make sure all blocks are lined up and even”).
>
> We further added new experiments in the supplementary material (Table 7, columns “Corr.”) to assess that our model can generalize to unseen instructions. Synthetic instructions were corrupted by replacing the color names seen during training with new color names (e.g. maroon becomes brown, cyan becomes light blue).
>
> - [R1] Finn, Chelsea, et al. "One-shot visual imitation learning via meta-learning." Conference on robot learning. PMLR, 2017.
> - [R2] James, Stephen, Michael Bloesch, and Andrew J. Davison. "Task-embedded control networks for few-shot imitation learning." Conference on robot learning. PMLR, 2018.
> - [R3] Jang, Eric, et al. "Bc-z: Zero-shot task generalization with robotic imitation learning." Conference on Robot Learning. PMLR, 2022.
> - [R4] Zheng, Kaizhi, et al. "VLMbench: A Compositional Benchmark for Vision-and-Language Manipulation." arXiv preprint arXiv:2206.08522 (2022).
>
> **The method for computing position offsets (Lines 198-201) doesn’t seem like it could handle cases where the same task at the same time-step may require a different offset based on the environment configuration. This reduces the generality of the approach. Furthermore, the assumption of a fixed set of tasks cements this as a multi-task learning approach rather than an instruction-driven approach. In instruction-driven work, it's questionable to map to a fixed set of tasks, because the set of tasks is by definition extensible via specification of new instructions.**
>
> The proposed way of predicting position offsets can be viewed as sharing predicted offsets for different variations of the same task. We use the offset to handle the cases where an agent should not directly reach an actual object in the scene. For example, in the “push button” task, the agent should first reach a target location above the button to avoid colliding it. Therefore, the proper range of offsets is similar for the same task under different variations at the same timestep, and thus empirically this design does not reduce the generality of our approach.
>
> To further improve the generalization of our approach to various instructions, we can simply adapt the model and use the feature map $H_t$ to predict the position offset. In this way, the offset is dynamic to the current environment configuration just like the prediction of position $p^e_t$. We achieved 76.8% (vs. 88.3% with our approach) with this model in the single-task setting, since it is harder to regress its value from $H_t$ than from the task logit.
>
> **No experiments on real robots, so tolerance to noise, or sim2real generalization couldn’t be assessed.**
>
> We provided experiments on real robots in the supplementary material (now moved to Section 5.4 in the main paper). We report results on a UR5 robotic arm for variations of the pushing buttons task. We first train our model using simulated demonstrations in the RLBench simulator, and then collect 10 real-world variations with 10 demonstrations per variation to finetune the model. We test on the real robot for both seen and unseen variations and with synthetic and human-written instructions. The experimental results show that our model achieves good sim2real generalization ability.

---

> > ### Author Response · Authors · 2022-08-27
> > **Answer to reviewer #1 (G4rm) [2/3]**
> >
> > **Why does applying “max-pooling” to point clouds make sense? Wouldn’t median-pooling (or even average-pooling) make more sense?**
> >
> > We use the point cloud features to provide positions relative to the base of the robot for each patch. Since the resolution of the encoded point cloud ($Pool(P_k)$) is quite low (8x8), we observe that the type of pooling has little influence on the performance. Compared to max pooling, mean pooling brings a slight improvement (+0.2%), whereas median pooling tends to slightly decrease performance (-0.1%). Due to time constraints during the rebuttal, we were not able to recompute all experiments using mean pooling, but we will update the description of the method and the results with mean pooling in the camera-ready version.
> >
> > **Why are you classifying instructions into a fixed set of task types?**
> >
> > There are two main objectives for classifying instructions into task types. Firstly, it serves as an auxiliary loss in training (Eq 8). Previous vision-and-language works [R5, R6] have shown that it is beneficial to have it as a regularizer.
> > Secondly, we use the predicted task types to predict position offsets (L196). As explained in the second reply, using the predicted task types to further predict the position offset improves the performance. Moreover, since the task type is a high-level categorization of instructions, it does allow the model to generalize to multi-variations with unseen instructions of a task.
> >
> > * [R5] Chen et al. "ScanRefer 3d object localization in rgb-d scans using natural language." ECCV, 2020
> > * [R6] Achlioptas et al. "Referit3d Neural listeners for fine-grained 3d object identification in real-world scenes." ECCV, 2020
> >
> > **Equation 6: what is LN? The inner layer was represented by explicitly writing down a matrix multiplication. Why does the outer layer use LN and what does it mean?**
> >
> > LN is layer norm and is widely used to normalize inputs passed to a transformer [71]. It is explained in Eq 1 (L148-149).
> >
> > **Equation 7: If the distribution is multi-modal, then averaging coordinates (e.g. points) may result in completely false predictions in-between the two modes, rather than better predictions on one of the two modes. Is this ever a problem in practice?**
> >
> > We did not observe this problem in practice. For example, we achieve a satisfactory success rate in the set of Multi-modal tasks in Table 3 (examples of such tasks is “pick up cup”), and qualitative examples in Figure 4 in the supplementary material also show that our model can deal with multi-modal cases.
> >
> > **Line 149 - what is $E_{T}^{x}$?**
> >
> > Tokens in multimodal transformers are usually computed as the sum of embeddings [R7, R8]: the image/word embedding, a positional embedding and a type embedding. A unique type embedding is used for visual and textual tokens, this  helps the Transformer model know  whether an input token is visual or textual. Here, $E_T^x$ is the type embedding.
> >
> > * [R7] Li et al. “VisualBERT: A Simple and Performant Baseline for Vision and Language”, ACL20.
> > * [R8] Tan et al. “LXMERT: Learning Cross-Modality Encoder Representations from Transformers”, IJCNLP 2019
> >
> > **Line 151 - maybe call A_{t}^{k} a “binary mask” instead of “attention map”? The word “attention” can have connotations of it being a soft mask, a mask that sums to 1, and a mask computed by a learned model for downstream use in the same or different learned model.**
> >
> > In the context of our work, the terminology “mask” is employed for randomly masking some tokens (see the masking procedure L204-208). An attention map is a 2D tensor that sums to 1. This is the case for  $A_t^k$ and $B_t^k$. $B_{t}^k$ is a soft-attention map, whereas $A_t^k$ is a hard-attention map. Therefore, $A_t^k$ is referred to as a “binary attention map”.
> >
> > **Line 181 - would be nice to have specific section numbers and links when referring to supplementary material.**
> >
> > Thank you for this suggestion. We added this to the revised version.
> >
> > **There are multiple prior works on instruction-following (e.g. [A, B, K, L]), that build maps as a means to provide a concise spatial memory to a robot. This is often seen as an advantage over keeping in memory a complete history of observations. Could the paper discuss why keeping history might be advantageous?**
> >
> > The above mentioned prior works focus on the task of navigation, where the spatial memory can be a topological map, a 2D image, or a 3D map to explicitly capture the explored environments.
> > It is still an open research problem whether such explicit maps are better than implicit memory [R10] and how to more effectively and efficiently store and process the explicit maps.
> > In the context of manipulation, the full history information has not been used in previous works. This work keeps history as a sequence and achieves SOTA performance. Adding explicit spatial memory into a history sequence could be evaluated in future work.
> >
> > [R10] Ruslan et al. "Is Mapping Necessary for Realistic PointGoal Navigation?" CVPR, 2022.

---

> > > ### Author Response · Authors · 2022-08-27
> > > **Answer to reviewer #1 (G4rm) [3/3]**
> > >
> > > **Section 4.2: is it fair to call this method a “multi-modal transformer”?**
> > >
> > > We used the term multimodal transformer as it is the de facto standard convention in prior transformer-based VLN works that take as input both vision and language tokens, see [77], [R11], [R12], [R13]. We will add a clarification to avoid confusion.
> > >
> > > [R11] Hubert et al. "Multimodal transformer for unaligned multimodal language sequences." ACL. 2019.
> > > [R12] Ronghang et al. "Unit: Multimodal multitask learning with a unified transformer." In ICCV. 2021.
> > > [R13] Wasifur et al. "Integrating multimodal information in large pretrained transformers." In ACL. 2020.

---

> > > > ### Comment · Reviewer_G4rm · 2022-08-27
> > > > **Thank you**
> > > >
> > > > Thank you for carefully addressing my concerns questions in detail! It clarifies a lot of my confusion.
> > > >
> > > > I can see how some of my concerns about generality can be easily addressed with slight tweaks to the computation.
> > > >
> > > > I like the paper, and I appreciate the significant amount of work it took to put it together, and would like to see it accepted!

---

### Author Response · Authors · 2022-08-27
**Update of the paper and the supplementary material**

**Comment:**

Accordingly to the discussions from the rebuttal, we have updated the paper (the attached PDF) and the supplementary material (the attached zip file). We have colorized modified text in blue.

**Zip File:**

/attachment/7b20cfddd1bb377b63ef3a43ed6dd0ecd3584cc7.zip

---

### Meta-Review · Area_Chair_XnT1 · 2022-08-14

**Recommendation:** Accept (Oral)
**Confidence:** 5

**Metareview:**

All reviewers agree that this paper is solid and well organized.
1. The paper shows superior performance in RLBench compared to reasonable baseline methods.
2. Although hardware experiments are not clearly mentioned in the main paper, supplemental video clearly demonstrates that the method has been successfully applied to a physical robot.

The reviewers raised some concerns:
1. In terms of instruction-driven manipulation, the vocabulary used in the task is very limited compared to other datasets (e.g. PFN-PIC). Linguistic information does not play a major role for the majority of the RLBench tasks.
2. The fact that hardware experiments are presented in the supplementary material is not clearly explained in the body of the paper.

### Post-rebuttal comment
The authors have sufficiently addressed most of the concerns. I recommend acceptance of the paper as "oral".


**Best Paper Nomination:**

No

---

> ### Author Response · Authors · 2022-08-27
> **Answer to the meta-reviewer.**
>
> We would like to thank our reviewers and our meta-reviewer for providing positive and constructive comments. We have updated the main paper and supplementary material following the reviewers’ comments, and we address the concerns in replies below.
>
> **In terms of instruction-driven manipulation, the vocabulary used in the task is very limited compared to other datasets (e.g. PFN-PIC). Linguistic information does not play a major role for the majority of the RLBench tasks.**
>
> We agree that the vocabulary used during training – provided by RLBench, is limited compared to the PFN-PIC dataset. However, the task of the PFN-PIC dataset is quite different, as their goal is to ground an object described in an instruction in static images or point clouds instead of performing actions with robots.
>
> While the instructions in RLBench are synthetic and simple, RLBench contains abundant tasks and variations of tasks with instructions which are comparable with or even more diverse than existing robotic manipulation benchmarks. Furthermore, apart from testing with synthetic instructions in RLBench, we collected human-written instructions. The variation of these human instructions is significant. For example, for the task “push the cyan button, push the gray button, push the green button”, a human-written instruction is “press the cyan, then move to the gray one and finish with the green button” or "press the darker blue button, then the grey one and finally the lighter blue button”. The results on the “push buttons” task (Table 4 Column Real and Table 7 in the supplementary material) showed that our model trained with simple synthetic instructions is able to generalize to natural instructions and to color names unseen during training. We also carried out real robot experiments with human-written instructions to demonstrate the effectiveness of our model in the real world.
>
> **The fact that hardware experiments are presented in the supplementary material is not clearly explained in the body of the paper**
>
> We agree and have moved the real robotic results to the main paper, please see the revised version.